# Retailer's Emergency Ordering Policy When Facing an Impending Supply Disruption

**Jingfu Huang** **, Gaoke Wu and Yiju Wang ***

School of Management Science, Qufu Normal University, Rizhao 276826, China; jingfu_huang@126.com (J.H.); wugaoke1@163.com (G.W.)
*   Correspondence: wyijumail@163.com; Tel.:+86-135-6330-9372

**Abstract:** Supply disruption is a common phenomenon in business activities. For the case where the supply disruption is predictable, the retailer should make an emergency procurement beforehand to decrease the inventory cost. For the scenario such that the happening time of the supply disruption obeys a certain common probability distribution but the ending time of the supply disruption is deterministic, based on minimizing the inventory cost and under two possible procurement strategies, we establish an emergency procurement optimization model. By considering the model solution in all cases, we establish a closed-form solution to the optimization model and provide an optimal emergency procurement policy to the retailer. Some numerical experiments are made to test the validity of the model and the effect of the involved parameters on the emergency procurement policy.

**Keywords:** supply disruption; emergency ordering policy; impending

## 1. Introduction

In the modern market, supply chain disruptions usually occur due to some unforeseen triggering events and jeopardize the flow of material and normal business activities significantly [1–8]. Generally, the risk of supply chain disruption comes from five distinct sections: demand, supply, regulatory, infrastructure, catastrophic [9]. Natural disruptive triggers include earthquakes, floods, fires, and man-made triggers include terrorist attacks, accidents, supplier bankruptcy, etc. [10]; well-documented disruption triggers include the 9/11 terrorist attacks, Hurricane Mitch, and the Taiwan earthquake in 1999. The resulting consequences of a disruption are dramatic. For example, the Taiwan earthquake in 1999 caused global supply disruption of the computer chip, the Philips fire in 2000 struck a fatal blow to Ericsson's mobile phone business, and a series of events such as the 9/11 incident in 2001 resulted in many industrial supply chain disruptions [11].

With the progress of globalization as well as outsourcing and an intensified focus on efficiency and lean management, the risk of supply chain disruptions has increased over the last decade [12], and the research of supply chain risk has attracted much attention from researchers [6,13–21].

To reduce the risk of supply chain disruption, the strategy of building resiliency into a supply chain is introduced in supply chain risk management. For this, Hendricks and Singhal [22] offered some suggestions for risk mitigation including improving forecast accuracy, synchronizing planning and execution, reducing lead times, collaboration with partners, and investing in technology. Hopp, Iravani, and Liu [23] detailed a plan that focuses on considering possible risks and how to prevent them, responding to disruptions when they occur, formulating plans that protect the supply chain, focusing on the overall supply chain structure, and planning for the process of recovering from possible disruptions. For large-scale disruptive events, Akkerman and Van Wassenhove [24] proposed a plan of sense making, decision making, implementation, and learning. Hendricks and Singhal [25] suggested that focusing on reducing the frequency and working to better predict disruptions will give businesses more time to resolve problems when they occur.

Meyer et al. [26] analyzed the reliability in meeting demand and the average inventory of the system in which the failure and repair processes of the production facility were random.

For a supply related disruption where the supply might be disrupted due to machine breakdown, Parlar [27] established an $(s, S)$-type optimal procurement policy. For the inventory mechanism with a stochastic demand and with a random product supply disruption for a period of random duration, Arreola-Risa and DeCroix [14] provided an optimal ordering policy to the retailer and provided a deep insight into the inventory mechanism. For the demand-related disruption, Qi, Bardb, and Yu [28] investigated the supply-chain-mechanism-derived conditions under which the supply chain can be coordinated for both centralized and decentralized decision-making. Xia et al. [29] considered disruption management for a two-stage production and inventory system with a linear or quadratic loss function, and obtained the optimal ordering policy. Xiao et al. [30] considered the coordination of a supply chain with one manufacturer and two competing retailers when the demand was disrupted; they showed that an appropriate contractual arrangement can fully coordinate the supply chain and the manufacturer can achieve a desired allocation of the total channel profit by varying the unit wholesale price and the subsidy rate. Huang, Yang, and Zhang [31] considered the scenario where the manufacturer and the retailer were vertically integrated with demand disruptions, and derived conditions under which the maximum profit can be achieved, which indicates that the optimal production quantity has some robustness under a demand disruption in both centralized and decentralized dual channel supply chains.

In this paper, we consider the emergency procurement mechanism with an impending supply disruption such that the happening time of the supply disruption is random but the ending time of the event is deterministic. This supply disruption mechanism is encountered in reality. For example, in September 2017, a vehicle parts maker, Schaeffler, in Asia was told that its only raw material supplier of needle roller was under an environmental inspection and supply of the raw material would be terminated within a month [32]. In this case, Schaeffler had to switch a new supplier but this would take at least three months. This means that the vehicle parts maker Schaeffler would face an impending supply disruption and the supply could only be resumed after three months.

For the concerned emergency procurement problem, to avoid out of stock and/or excessive procurement, and thus decrease the inventory cost, the retailer should make an emergency procurement in advance. However, if the emergency order is placed too early, then the retailer should bear more inventory cost, and if it is made too late, then he may lose the chance to place an emergency order. This means that the retailer should determine the emergency ordering time and emergency ordering quantity. For this problem, based on minimizing the retailer's inventory cost, we establish an emergency procurement optimization model. By the model solution, in all cases, we obtain a closed-form solution to the optimization model and provide an optimal emergency procurement policy to the retailer.

The remainder of the paper is organized as follows. Section 2 gives the notations and assumptions needed on the concerned problem. Section 3 establishes an optimization model for the concerned problem. Section 4 gives the closed form solution of the optimization model and provide an optimal emergency procurement policy to the retailer. Some numerical experiments are made in Section 5 to test the effect of the involved parameters on the model, which helps the retailer to focus attention on the crucial parameters when making the emergency order. Some conclusions are drawn in the last section.

## 2. Notations and Assumptions

For the emergency procurement problem with an impending supply disruption, we assume that the demand is stable, the time when the supply disruption occurs is random, and the ending time of the event is deterministic. In particular, we assume that the happening time of the supply disruption obeys a certain common probability distributions

in $[0, t_1]$ and the ending time of the event is $t_2$. This means that the deadline of the event is $t_1$ and the supply can be recovered at $t_2$. For the emergency procurement problem, to prevent being out of stock and hence decreasing the cost of being out of stock, the retailer should make an emergency order before $t_1$. For convenience, we denote the happening time of the event by $t_p$ and denote the remaining inventory at the start time by $Q_0$. Then, the retailer should make an emergency order at $t_e$ (variable) with quantity $Q_e$ (variable) before $t_1$ according to the demand during the supply disruption and the remaining inventory $Q_0$, see Figure 1. However, since the happening time of the supply disruption is random, if the retailer makes the emergency order too early, the retailer will bear a high inventory holding cost. Thus, the retailer may first make a regular order with quantity $Q_r$ at the time when the remaining inventory $Q_0$ is depleted, then make an emergency order with quantity $Q_e$ when the regular order inventory is depleted, see Figure 2. In this sense, there are two procurement strategies for the concerned problem. To make the inventory cost as low as possible, the retailer should determine the optimal emergency ordering time and ordering quantity. To this end, we need the following notations, see Table 1.

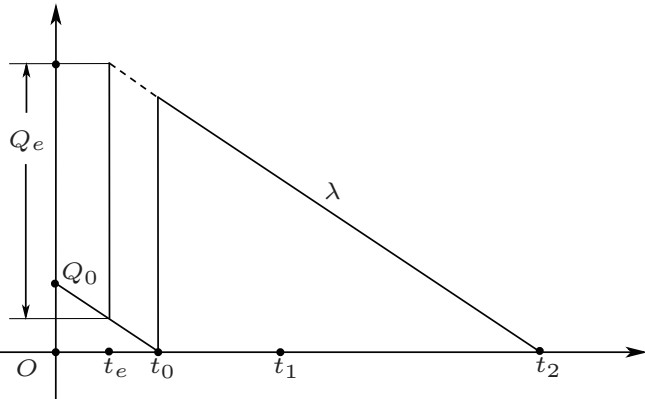

**Figure 1.** Order before $Q_0$ is depleted.

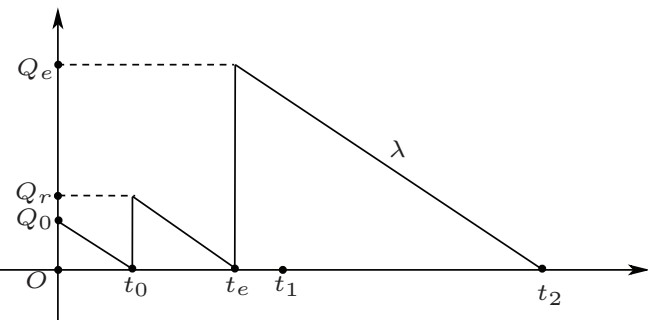

**Figure 2.** Order after $Q_0$ is depleted.

**Table 1.** Notations.

| Symbol | Description | Symbol | Description |
|--------|-------------|--------|-------------|
| $K$ | fixed order cost | $t_p$ | happening time of supply disruption |
| $\lambda$ | demand rate | $t_2$ | ending time of supply disruption |
| $h$ | holding cost per time per unit | $Q_0$ | remaining inventory at the happening time |
| $p$ | out of stock cost per unit | $Q_r$ | regular order quantity before emergency order(variable) |
| $t_0$ | the time when $Q_0$ is depleted | $Q_e$ | emergency order quantity (variable) |
| $t_1$ | the deadline happening time of supply disruption | $t_e$ | emergency ordering time (variable) |
| | | $*$ | indicates the optimal value |

For the concerned model, we further make the following assumptions:

**Assumption 1.** *(1) The demand throughout the system is stable with rate $\lambda$;*
*(2) The happening time of the supply chain disruption obeys the uniform distribution in $[0, t_1]$ or*
   *the distribution such that the probability of occurrence is linearly increasing in $[0, t_1]$;*
*(3) The retailer only makes one emergency order before $t_1$.*

For the assumption on the probability distribution of the supply disruption happening time, i.e., Assumption (2) given above, it is based on the following considerations: the supply disruption may occur with an equal probability in $[0, t_1]$ or the probability of the event in $[0, t_1]$ increases gradually over time. The probability density function of the happening time of the supply disruption is $f(x) = \frac{1}{t_1}$ with $x \in [0, t_1]$ for the former case, and the probability density function is $f_2(x) = \frac{2x}{t_1^2}$ with $x \in [0, t_1]$ for the latter case.

## 3. Mathematical Formulation

For the concerned inventory mechanism, as the market demand rate is $\lambda$, the retailer's remaining inventory $Q_0$ at the beginning will be depleted at $\frac{Q_0}{\lambda}$. Hence, if the retailer makes an emergency order before $\frac{Q_0}{\lambda}$, then the retailer's emergency order quantity is $Q_e = (\lambda t_2 - Q_0)$, and if the emergency order is made after $\frac{Q_0}{\lambda}$, then the retailer should first make a regular order with quantity $Q_r$ at $\frac{Q_0}{\lambda}$ and make an emergency order with quantity $Q_e = (\lambda t_2 - Q_0 - Q_r)$ at $\frac{Q_0 + Q_r}{\lambda}$. Thus, the discussion on the emergency procurement problem can be broken into two cases.

**Case I:** The retailer makes an emergency order before $\frac{Q_0}{\lambda}$. In this case, if the supply disruption occurs after the emergency order, then the retailer's inventory cost is

$$C_1^1(t_e) = \frac{hQ_0^2}{2\lambda} + K + \frac{h(\lambda t_2 - Q_0)^2}{2\lambda} + h(\lambda t_2 - Q_0)(\frac{Q_0}{\lambda} - t_e),$$

where the first term is the holding cost of the remaining inventory $Q_0$ at beginning, and the last three terms are inventory cost for the emergency order [33].

Certainly, if the supply disruption happens before the emergency order, i.e., $t_p < t_e$, then the retailer would miss the chance of making an emergency order, and the retailer's inventory cost is

$$C_2^1(t_e) = \frac{hQ_0^2}{2\lambda} + p(\lambda t_2 - Q_0),$$

where the first term is the holding cost of the remaining inventory at the beginning, and the second term is the penalty cost of being out of stock.

For the case that the happening time of the supply disruption obeys the uniform distribution in $[0, t_1]$, the retailer's expected inventory cost for this emergency procurement policy is

$$\begin{aligned}
F_1(t_e) &= \int_{t_e}^{t_1} C_1^1(t_e) f_1(x) dx + \int_1^{t_e} C_1^2(t_e) f_1(x) dx \\
&= \left(1 - \frac{t_e}{t_0}\right) \left(\frac{hQ_0^2}{2\lambda} + K + \frac{h(\lambda t_2 - Q_0)^2}{2\lambda} + h(\lambda t_2 - Q_0)(\frac{Q_0}{\lambda} - t_e)\right) \\
&\quad + \frac{t_e}{t_0}\left(\frac{hQ_0^2}{2\lambda} + p(\lambda t_2 - Q_0)\right) \\
&= \frac{t_e^2}{t_0} h(\lambda t_2 - Q_0) - \frac{t_e}{t_0}\left(K + \frac{h(\lambda t_2 - Q_0)^2}{2\lambda} + (\lambda t_2 - Q_0)\frac{hQ_0}{\lambda}\right. \\
&\quad \left. - p(\lambda t_2 - Q_0) + t_1 h(\lambda t_e - Q_0)\right) \\
&\quad + \frac{hQ_0^2}{2\lambda} + K + \frac{h(\lambda t_2 - Q_0)^2}{2\lambda} + h(\lambda t_2 - Q_0)\frac{Q_0}{\lambda}.
\end{aligned}$$

Since $\lambda t_2 - Q_0 > 0$, the function is a quadratic convex function in $t_e$.

For the case that the probability of the happening time of the event increases gradually over time in $[0, t_1]$, the retailer's expected inventory cost for this emergency procurement policy is

$$G_1(t_e) = \int_{t_e}^{t_0} C_1^1(t_e) f_2(x) dx + \int_0^{t_e} C_1^2(t_e) f_2(x) dx$$

$$= \left(1 - \frac{t_e^2}{t_1^2}\right) \left(\frac{hQ_0^2}{2\lambda} + K + \frac{h(\lambda t_2 - Q_0)^2}{2\lambda} + h(\lambda t_2 - Q_0)(\frac{Q_0}{\lambda} - t_e)\right)$$

$$+ \frac{t_e^2}{t_1^2} \left(\frac{hQ_0^2}{2\lambda} + p(\lambda t_2 - Q_0)\right)$$

$$= \frac{t_e^3}{t_0^2} h(\lambda t_2 - Q_0) - \frac{t_e^2}{t_0^2}\left(K + \frac{h(\lambda t_2 - Q_0)^2}{2\lambda} + (\lambda t_2 - Q_0)\frac{hQ_0}{\lambda} - p(\lambda t_2 - Q_0)\right)$$

$$+ t_e h(\lambda t_2 - Q_0) + \frac{hQ_0^2}{2\lambda} + K + \frac{h(\lambda t_2 - Q_0)^2}{2\lambda} + h(\lambda t_2 - Q_0)\frac{Q_0}{\lambda},$$

which is a cubic function of $t_e$.

**Case II:** The retailer makes an emergency order after $\frac{Q_0}{\lambda}$, i.e., $\frac{Q_0}{\lambda} < t_e$. In this case, the retailer first makes a regular order at $\frac{Q_0}{\lambda}$ with quantity $Q_r \in [0, \lambda t_1 - Q_0]$ and makes an emergency order at $t_e = \frac{Q_0 + Q_r}{\lambda}$ with quantity $\lambda t_2 - Q_0 - Q_r$. This means that once the regular procurement policy is determined, the emergency procurement policy is also determined.

For this case, if the supply disruption occurs after the emergency order, i.e., $t_e \leq t_p$, then the retailer's inventory cost is

$$C_1^2(t_e, Q_r) = \frac{hQ_0^2}{2\lambda} + K + \frac{hQ_r^2}{2\lambda} + K + \frac{h(\lambda t_2 - Q_0 - Q_r)^2}{2\lambda}$$

$$= 2K + \frac{h(Q_0^2 + Q_r^2)}{2\lambda} + \frac{h(\lambda t_2 - Q_0 - Q_r)^2}{2\lambda},$$

and if the emergency order is made after the disruption happening time, i.e., $t_p < t_e$, then the retailer would miss the optimal procurement chance, and the retailer's inventory cost is

$$C_2^2(t_e, Q_r) = \frac{hQ_0^2}{2\lambda} + K + \frac{hQ_r^2}{2\lambda} + p(\lambda t_2 - Q_0 - Q_r)$$

$$= K + \frac{h(Q_0^2 + Q_r^2)}{2\lambda} + p(\lambda t_2 - Q_0 - Q_r).$$

If the happening time of the event obeys the uniform distribution in $[0, t_1]$, then the retailer's expected inventory cost for this replenishment policy is

$$F_2(Q_r) = \int_{t_e}^{t_0} C_1^2(t_e, Q_r) f_1(x) dx + \int_0^{t_e} C_2^2(t_e, Q_r) f_1(x) dx$$

$$= \left(1 - \frac{t_e}{t_1}\right)\left(2K + \frac{h(Q_0^2 + Q_r^2)}{2\lambda} + \frac{h(\lambda t_2 - Q_0 - Q_r)^2}{2\lambda}\right)$$

$$+ \frac{t_e}{t_0}\left(K + \frac{h(Q_0^2 + Q_r^2)}{2\lambda} + p(\lambda t_2 - Q_0 - Q_r)\right)$$

$$= \frac{t_1\lambda - Q_0 - Q_r}{t_1\lambda}\left(2K + \frac{h(Q_0^2 + Q_r^2)}{2\lambda} + \frac{h(\lambda t_2 - Q_0 - Q_r)^2}{2\lambda}\right)$$

$$+ \frac{Q_0 + Q_r}{t_1\lambda}\left(K + \frac{h(Q_0^2 + Q_r^2)}{2\lambda} + p(\lambda t_2 - Q_0 - Q_r)\right)$$

$$= \left(2K + \frac{h(Q_0^2 + Q_r^2)}{2\lambda} + \frac{h(\lambda t_2 - Q_0 - Q_r)^2}{2\lambda}\right)$$

$$- \frac{Q_0 + Q_r}{t_1\lambda}\left(K + \frac{h(\lambda t_2 - Q_0 - Q_r)^2}{2\lambda} - p(\lambda t_2 - Q_0 - Q_r)\right),$$

which is a cubic function of $Q_r$.

If the happening time of the supply disruption obeys the distribution with probability density function $f_2(x) = \frac{2x}{t_1^2}$, $x \in [0, t_1]$, then the retailer's expected inventory cost for this replenishment policy is

$$G_2(Q_r) = \int_{t_e}^{t_1} C_1^2(t_e, Q_r) f_2(x) dx + \int_0^{t_e} C_2^2(t_e, Q_r) f_2(x) dx$$

$$= \left(1 - \frac{t_e^2}{t_1^2}\right)\left(2K + \frac{h(Q_0^2 + Q_r^2)}{2\lambda} + \frac{h(\lambda t_2 - Q_0 - Q_r)^2}{2\lambda}\right)$$

$$+ \frac{t_e^2}{t_1^2}\left(K + \frac{h(Q_0^2 + Q_r^2)}{2\lambda} + p(\lambda t_2 - Q_0 - Q_r)\right)$$

$$= \frac{t_1^2 \lambda^2 - (Q_0 + Q_r)^2}{t_0^2 \lambda^2}\left(2K + \frac{h(Q_0^2 + Q_r^2)}{2\lambda} + \frac{h(\lambda t_2 - Q_0 - Q_r)^2}{2\lambda}\right)$$

$$+ \frac{(Q_0 + Q_r)^2}{t_1^2 \lambda^2}\left(K + \frac{h(Q_0^2 + Q_r^2)}{2\lambda} + p(\lambda t_2 - Q_0 - Q_r)\right)$$

$$= -\frac{h(Q_0 + Q_r)^2(\lambda t_2 - Q_0 - Q_r)^2}{2 t_1^2 \lambda^3} + \frac{(p(\lambda t_2 - Q_0 - Q_r) - K)(Q_0 + Q_r)^2}{t_0^2 \lambda^2}$$

$$+ \frac{h(\lambda t_2 - Q_0 - Q_r)^2}{2\lambda} + \frac{h(Q_0^2 + Q_r^2)}{2\lambda} + 2K,$$

which is a quartic function of $Q_r$.

Based on the discussion above, if the happening time of the supply chain disruption obeys the uniform distribution in $[0, t_1]$, then the concerned problem can be formulated as the following optimization problem

$$\begin{aligned} \min \quad & \{F_1(t_e), F_2(Q_r)\} \\ \text{s.t.} \quad & 0 \le t_e \le t_1 \\ & 0 \le Q_r \le \lambda t_1 - Q_0, \end{aligned} \tag{1}$$

and if the happening time of the event obeys the distribution with probability density function $f_2(x) = \frac{2x}{t_1^2}$ in $[0, t_1]$, then the concerned problem can be formulated as the following optimization problem:

$$\begin{aligned} \min \quad & \{G_1(t_e), G_2(Q_r)\} \\ \text{s.t.} \quad & 0 \le t_e \le t_1 \\ & 0 \le Q_r \le \lambda t_1 - Q_0. \end{aligned} \tag{2}$$

In the next section, we will establish its closed-form solution by analyzing the model.

## 4. Model Solution

To solve the emergency procurement optimization models established in Section 3, we break the discussion into two cases according to the distribution of happening time of the supply disruption.

If the happening time of the supply disruption obeys the uniform distribution, then we need to solve the optimization problem (1). For this, if the emergency order is made before $\frac{Q_0}{\lambda}$, then we have the following conclusion.

**Theorem 1.** *If the supply chain disruption happening time obeys the uniform distribution and the emergency order is made before time $\frac{Q_0}{\lambda}$, then the optimal emergency ordering time is*

$$t_e^* = \text{med}\left\{0, \frac{t_1}{2} + \frac{t_2}{4} - \frac{p}{2h} + \frac{Q_0}{4\lambda} + \frac{K}{2h(\lambda t_2 - Q_0)}, \frac{Q_0}{\lambda}\right\}.$$

**Proof.** From the assumption, the retailer's expected inventory cost is

$$\begin{aligned} F_1(t_e) =& \frac{h(\lambda t_2 - Q_0)}{t_1} t_e^2 + \left(\frac{p(\lambda t_2 - Q_0)}{t_1} - \frac{h(\lambda^2 t_2^2 - Q_0^2)}{2\lambda t_1} - \frac{K}{t_1}\right) t_e \\ & - h(\lambda t_2 - Q_0) t_e + \frac{h \lambda t_2^2}{2} + K. \end{aligned}$$

It is easy to compute that

$$F_1'(t_e) = \frac{2h(\lambda t_2 - Q_0)}{t_1} t_e + \frac{2p\lambda(\lambda t_2 - Q_0) - h(\lambda^2 t_2^2 - Q_0^2) - 2\lambda K}{2\lambda t_1} - h(\lambda t_2 - Q_0).$$

Since

$$F_1''(t_e) = \frac{2h(\lambda t_2 - Q_0)}{t_1} > 0,$$

the minimum of function $F_1(t_e)$ is the root of equation $F_1'(t_e) = 0$, i.e.,

$$\bar{t}_e = \frac{t_1}{2} + \frac{t_2}{4} - \frac{p}{2h} + \frac{Q_0}{4\lambda} + \frac{K}{2h(\lambda t_2 - Q_0)}.$$

Taking the constraints $0 \le t_e \le \frac{Q_0}{\lambda}$ into consideration, we can obtain the retailer's optimal emergency ordering time $\text{med}\{0, \bar{t}_e, \frac{Q_0}{\lambda}\}$ and the desired result follows. $\quad\square$

For the case that the emergency order is made after the time $\frac{Q_0}{\lambda}$, we have the following conclusion.

**Theorem 2.** *If the supply chain disruption happening time obeys the uniform distribution and the emergency order is made after the time $\frac{Q_0}{\lambda}$, then the optimal regular order quantity is*

$$Q_r^* = \begin{cases} \max(Q_r^+, 0), & \text{if } b^2 - 4ac > 0, Q_r^+ \le \lambda t_1 - Q_0, F_2(\max(Q_r^+, 0)) \le F_2(\lambda t_1 - Q_0); \\ \lambda t_1 - Q_0, & \text{otherwise,} \end{cases}$$

*where*

$$Q_r^+ = (-b + \sqrt{b^2 - 4ac})/(2a)$$

*and*

$$\begin{cases} a = \dfrac{-3h}{2t_1\lambda^2}, & b = (4h\lambda t_1 + 4h\lambda t_2 - 4p\lambda - 6hQ_0)/(2t_1\lambda^2), \\ c = (-2h\lambda^2 t_1 t_2 - h\lambda^2 t_2^2 - 3hQ_0^2 + 4h\lambda t_2 Q_0 + 2h\lambda t_1 Q_0 + 2p\lambda^2 t_2 - 4p\lambda Q_0 - 2\lambda K)/(2t_1\lambda^2). \end{cases}$$

*The emergency order is made at $t_e^* = \frac{Q_0 + Q_r^*}{\lambda}$ with quantity $Q_e^* = \lambda t_2 - Q_0 - Q_r^*$.*

**Proof.** Since the emergency order is made after time $\frac{Q_0}{\lambda}$, from the assumption and the discussion in Section 3, the retailer's expected inventory cost is

$$F_2(Q_r) = \left(2K + \frac{h(Q_0^2 + Q_r^2)}{2\lambda} + \frac{h(\lambda t_2 - Q_0 - Q_r)^2}{2\lambda}\right)$$
$$- \frac{Q_0 + Q_r}{t_1\lambda}\left(K + \frac{h(\lambda t_2 - Q_0 - Q_r)^2}{2\lambda} - p(\lambda t_2 - Q_0 - Q_r)\right).$$

To obtain its minimal point, we consider its critical points by computing its derivative.

$$F_2'(Q_r) = \left(\frac{hQ_r}{\lambda} - \frac{h(\lambda t_2 - Q_0 - Q_r)}{\lambda}\right) - \frac{1}{t_1\lambda}\left(K + \frac{h(\lambda t_2 - Q_0 - Q_r)^2}{2\lambda} - p(\lambda t_2 - Q_0 - Q_r)\right)$$
$$- \frac{Q_0 + Q_r}{t_1\lambda}\left(-\frac{h(\lambda t_2 - Q_0 - Q_r)}{\lambda} + p\right)$$
$$= \frac{2hQ_r - h\lambda t_2 + hQ_0}{\lambda} - \frac{1}{t_1\lambda}\left(K + \frac{ht_2(\lambda t_2 - Q_0 - Q_r)}{2} - p\lambda t_2 + 2p(Q_0 + Q_r)\right.$$
$$\left. - \frac{3h(Q_0 + Q_r)(\lambda t_2 - Q_0 - Q_r)}{2\lambda}\right)$$
$$= \frac{1}{2t_1\lambda^2}\left(-3hQ_r^2 + (4h\lambda t_1 + 4h\lambda t_2 - 4p\lambda - 6hQ_0)Q_r\right.$$
$$\left. - 2h\lambda^2 t_1 t_2 - h\lambda^2 t_2^2 - 3hQ_0^2 + 4h\lambda t_2 Q_0 + 2h\lambda t_1 Q_0 + 2p\lambda^2 t_2 - 4p\lambda Q_0 - 2\lambda K\right).$$

Certainly, the equation $F_2'(Q_r) = 0$ has two roots

$$Q_r^+ = \frac{-b + \sqrt{b^2 - 4ac}}{2a}, \quad Q_r^- = \frac{-b - \sqrt{b^2 - 4ac}}{2a},$$

where $a, b, c$ are that given in the assertion. Since the coefficient of the highest item of variable $Q_r$ in the function is negative, it holds that $\lim_{Q_r \to +\infty} F_2(Q_r) = -\infty$ and $\lim_{Q_r \to -\infty} F_2(Q_r) = +\infty$. Then, we can break the discussion into two cases.

**Case 1.** $b^2 - 4ac > 0$. In this case, function $F_2(Q_r)$ has two critical points $Q_r^+$ and $Q_r^-$ with $Q_r^+ < Q_r^-$. Further, the function $F_2(Q_r)$ is monotonically decreasing in $(-\infty, Q_r^+]$ and $[Q_r^-, +\infty)$, and monotonically increasing in $[Q_r^+, Q_r^-]$. Then, we have the following replenishment strategy for this case:

$$Q_r^* = \begin{cases} 0, & \text{if } Q_r^+ < 0, F_2(0) \le F_2(\lambda t_1 - Q_0), \\ \lambda t_1 - Q_0, & \text{if } Q_r^+ < 0, F_2(0) > F_2(\lambda t_1 - Q_0), \\ Q_r^+, & \text{if } 0 \le Q_r^+ \le \lambda t_1 - Q_0, F_2(Q_r^+) \le F_2(\lambda t_1 - Q_0), \\ \lambda t_1 - Q_0, & \text{if } 0 \le Q_r^+ \le \lambda t_1 - Q_0, F_2(Q_r^+) > F_2(\lambda t_1 - Q_0), \\ \lambda t_1 - Q_0, & \text{if } Q_1^+ > \lambda t_1 - Q_0. \end{cases}$$

**Case 2.** $b^2 - 4ac \le 0$. In this case, function $F_2(Q_r)$ is monotonically decreasing in $(-\infty, +\infty)$. Thus, the minimum point of function $F_2(Q_1)$ in $[0, \lambda t_1 - Q_0]$ is $\lambda t_1 - Q_0$, and the optimal regular ordering quantity is $Q_r^* = \lambda t_1 - Q_0$.

Combining the conclusions obtained above, we can obtain the desired result. $\quad\square$

From Theorems 1 and 2, we can obtain Algorithm 1 for solving optimization model (1).

---

**Algorithm 1:** Solution method for optimization model (1).

---

**Step 1.** Input parameters: $\lambda, K, h, p, Q_0, t_1, t_2$.

**Step 2.** Compute $\bar{t}_e = \text{med}\left\{0, \frac{t_1}{2} + \frac{t_2}{4} - \frac{p}{2h} + \frac{Q_0}{4\lambda} + \frac{K}{2h(\lambda t_2 - Q_0)}, \frac{Q_0}{\lambda}\right\}$.

**Step 3.** Set

$$\begin{cases} a = \dfrac{-3h}{2t_1\lambda^2}, \quad b = \dfrac{4h\lambda t_1 + 4h\lambda t_2 - 4p\lambda - 6hQ_0}{2t_1\lambda^2}, \\ c = \dfrac{-2h\lambda^2 t_1 t_2 - h\lambda^2 t_2^2 - 3hQ_0^2 + 4h\lambda t_2 Q_0 + 2h\lambda t_1 Q_0 + 2p\lambda^2 t_2 - 4p\lambda Q_0 - 2\lambda K}{2t_1\lambda^2}. \end{cases}$$

If $b^2 - 4ac > 0$, then set $\bar{Q}_r = \frac{-b + \sqrt{b^2 - 4ac}}{2a}$ and go to Step 4, else set $Q_r = \lambda t_1 - Q_0$ and go to Step 5.

**Step 4.** If $\bar{Q}_r < 0$ or $\bar{Q}_r > \lambda t_1 - Q_0$, then $Q_r = 0$, go to next step.

**Step 5.** If $F_2(Q_r) \le F_2(\lambda t_1 - Q_0)$, then set $Q_r^* = Q_r$, else $Q_r^* = \lambda t_1 - Q_0$.

**Step 6.** If $F_1(\bar{t}_e) \le F_2(Q_r^*)$, then $t_e^* = \bar{t}_e$, else $t_e^* = \frac{Q_0 + Q_r^*}{\lambda}$.

**Step 7.** If $t_e^* \le \frac{Q_0}{\lambda}$, then $Q_r^* = \lambda t_2 - Q_0$, else $Q_r^* = \lambda t_2 - Q_r^* - Q_0$.

**Step 8.** Output $t_e^*, Q_r^*, Q_e^*$.

---

From Theorems 1 and 2, we conclude that the algorithm can output a global optimal solution to problem (1).

Now, we consider the case that the happening time of the supply disruption obeys the distribution with probability density function $f_2(x) = \frac{2x}{t_1^2}$ $x \in [0, t_1]$. That is, we need to solve the optimization model (2). For the model, if the emergency order is made before $\frac{Q_0}{\lambda}$, we have the following conclusion.

**Theorem 3.** *If the supply chain disruption happening time obeys the distribution with probability density function $f_2(x) = \frac{2x}{t_1^2}$ $x \in [0, t_1]$ and the emergency order is made before the time $\frac{Q_0}{\lambda}$, then the optimal emergency ordering time is*

$$t_e^* = \begin{cases} \min(t_e^+, \frac{Q_0}{\lambda}), & \text{if } b_1^2 - 4a_1c_1 > 0, \quad G_1(0) \geq G_1(\min(t_e^+, \frac{Q_0}{\lambda})); \\ 0, & \text{otherwise.} \end{cases}$$

*where $t_e^+ = \frac{-b_1 + \sqrt{b_1^2 - 4a_1c_1}}{2a_1}$ and*

$$\begin{cases} a_1 = \dfrac{3h(\lambda t_2 - Q_0)}{t_1^2}, \quad c_1 = h(\lambda t_2 - Q_0), \\ b_1 = \dfrac{2p\lambda(\lambda t_2 - Q_0) - h(\lambda t_2 - Q_0)^2 - 2h(\lambda t_2 - Q_0)Q_0 - 2\lambda K}{\lambda t_1^2}. \end{cases}$$

**Proof.** From the assumption and the discussion in Section 3, the retailer's expected inventory cost is

$$G_1(t_e) = \frac{t_e^3}{t_1^2} h(\lambda t_2 - Q_0) - \frac{t_e^2}{t_1^2} \left( K + \frac{h(\lambda t_2 - Q_0)^2}{2\lambda} + (\lambda t_2 - Q_0)\frac{hQ_0}{\lambda} - p(\lambda t_2 - Q_0) \right)$$
$$+ t_e h(\lambda t_2 - Q_0) + \frac{hQ_0^2}{2\lambda} + K + \frac{h(\lambda t_2 - Q_0)^2}{2\lambda} + h(\lambda t_2 - Q_0)\frac{Q_0}{\lambda},$$

which is a cubic function of $t_e$. To compute its minimum, we compute its critical points by considering its derivative.

$$G_1'(t_e) = \frac{3h(\lambda t_2 - Q_0)}{t_1^2} t_e^2 + \left( \frac{2p(\lambda t_2 - Q_0)}{t_1^2} - \frac{h(\lambda t_2 - Q_0)^2}{\lambda t_1^2} - \frac{2h(\lambda t_2 - Q_0)Q_0}{\lambda t_1^2} - \frac{2K}{t_1^2} \right) t_e$$
$$+ h(\lambda t_2 - Q_0).$$

Certainly, the equation $G_1'(t_e) = 0$ has at most two roots

$$t_e^- = \frac{-b_1 - \sqrt{b_1^2 - 4a_1c_1}}{2a_1}, \quad t_e^+ = \frac{-b_1 + \sqrt{b_1^2 - 4a_1c_1}}{2a_1},$$

where $a_1, b_1, c_1$ are that given in the assertion. Since the coefficient of the highest item of variable $t_e$ in the function is positive, it holds that $\lim_{t_e \to +\infty} G_1(t_e) = +\infty$ and $\lim_{t_e \to -\infty} G_1(t_e) = -\infty$. Then, we can break the discussion into two cases.

**Case 1.** $b_1^2 - 4a_1c_1 > 0$. This function has two critical points

$$t_e^- = \frac{-b_1 - \sqrt{b_1^2 - 4a_1c_1}}{2a_1}, \quad t_e^+ = \frac{-b_1 + \sqrt{b_1^2 - 4a_1c_1}}{2a_1}.$$

In this case, the function $G_1(t_e)$ is monotonically decreasing in $[t_e^-, t_e^+]$ and monotonically increasing in $(-\infty, t_e^-]$ and $[t_e^+, +\infty)$. Then, the optimal emergency ordering time is

$$t_e^* = \begin{cases} 0, & \text{if } t_e^+ < 0, \\ 0, & \text{if } 0 \leq t_e^+ \leq \frac{Q_0}{\lambda}, G_1(0) \leq G_1(t_e^+), \\ t_e^+, & \text{if } 0 \leq t_e^+ \leq \frac{Q_0}{\lambda}, G_1(0) > G_1(t_e^+), \\ 0, & \text{if } t_e^+ > \frac{Q_0}{\lambda}, G_1(0) \leq G_1(\frac{Q_0}{\lambda}), \\ \frac{Q_0}{\lambda}, & \text{if } t_e^+ > \frac{Q_0}{\lambda}, G_1(0) > G_1(\frac{Q_0}{\lambda}). \end{cases}$$

**Case 2.** $b_1^2 - 4a_1c_1 \leq 0$. In this case, as $a_1 > 0$, $G_1'(t_e) \geq 0$, it implies that the function $G_1(t_e)$ is monotonically increasing in $(-\infty, +\infty)$. Thus, the minimum of the $G_1(t_e)$ in $[0, \lambda t_1 - Q_0]$ is reached at 0.

Combining the conclusions obtained above, we can obtain the desired assertion. □

Now, we consider the model solution of optimization problem (2) for the case that the emergency order is made after $\frac{Q_0}{\lambda}$. To this end, we need the following conclusion given in [34].

**Lemma 1.** *For a cubic equation* $ax^3 + bx^2 + cx + d = 0$ *with* $a > 0$, *set* $A = b^2 - 3ac$, $B = bc - 9ad$, $C = c^2 - 3bd$. *If* $A = B = 0$, *then the equation has a triple real root:* $x_1 = x_2 = x_3 = -\frac{c}{b}$; *if* $\Delta = B^2 - 4AC > 0$, *then the equation has one real root:* $x = \frac{-b - \left(\sqrt[3]{Y_1} + \sqrt[3]{Y_2}\right)}{3a}$, *where* $Y_1 = Ab + \frac{3a}{2}\left(-B + \sqrt{B^2 - 4AC}\right)$, $Y_2 = Ab + \frac{3a}{2}\left(-B - \sqrt{B^2 - 4AC}\right)$; *if* $\Delta = B^2 - 4AC = 0$, *then it has two real roots* $x_1 = -\frac{b}{a} + \frac{B}{A}$, $x_2 = x_3 = -\frac{B}{2A}$; *and if* $\Delta = B^2 - 4AC < 0$, *then it has three real roots* $x_1 = -\frac{1}{3a}(b + 2\sqrt{A}\cos\frac{\theta}{3})$, $x_2 = \frac{1}{3a}(-b + \sqrt{A}(\cos\frac{\theta}{3} + \sqrt{3}\sin\frac{\theta}{3}))$, $x_3 = \frac{1}{3a}(-b + \sqrt{A}(\cos\frac{\theta}{3} - \sqrt{3}\sin\frac{\theta}{3}))$, *where* $\theta = \arccos(\frac{2Ab - 3aB}{2\sqrt{A^3}})$.

**Theorem 4.** *If the supply chain disruption happening time obeys the distribution with the probability density function* $f_2(x) = \frac{2x}{t_1^2}$ $x \in [0, t_1]$ *and the emergency order is made after the time* $\frac{Q_0}{\lambda}$, *then the optimal regular ordering quantity* $Q_r^*$ *is*

$$Q_r^* = \begin{cases} 0, & \text{if } \Delta < 0, \bar{Q}_1 < 0, G_2(0) \leq G_2(\lambda t_1 - Q_0); \\ \lambda t_1 - Q_0, & \text{if } \Delta < 0, \bar{Q}_r < 0, G_2(0) > G_2(\lambda t_1 - Q_0); \\ 0, & \text{if } \Delta < 0, 0 \leq \bar{Q}_r \leq \lambda t_1 - Q_0, G_2(0) \leq \min(G_2(\bar{Q}_r), G_2(\lambda t_1 - Q_0)); \\ \bar{Q}_r, & \text{if } \Delta < 0, 0 \leq \bar{Q}_r \leq \lambda t_1 - Q_0, G_2(\bar{Q}_r) \leq \min(G_2(0), G_2(\lambda t_1 - Q_0)); \\ \lambda t_1 - Q_0, & \text{if } \Delta < 0, \bar{Q}_r < 0, G_2(\lambda t_1 - Q_0) \leq \min(G_2(0), G_2(\bar{Q}_r)); \\ 0, & \text{if } \Delta < 0, \bar{Q}_r > \lambda t_1 - Q_0, G_2(0) \leq G_2(\lambda t_1 - Q_0); \\ \lambda t_1 - Q_0, & \text{if } \Delta < 0, \bar{Q}_r > \lambda t_1 - Q_0, G_2(0) > G_2(\lambda t_1 - Q_0); \\ 0, & \text{if } \Delta \geq 0, \quad G_2(0) \leq G_2(\lambda t_1 - Q_0); \\ \lambda t_1 - Q_0. & \text{if } \Delta \geq 0, \quad G_2(0) > G_2(\lambda t_1 - Q_0), \end{cases}$$

*where* $\bar{Q}_r = \frac{1}{3a_2}(-b_2 + \sqrt{A}(\cos\frac{\theta}{3} + \sqrt{3}\sin\frac{\theta}{3}))$ *and*

$$\begin{cases} a_2 = \dfrac{2h}{t_0^2\lambda^3}, \quad b_2 = \dfrac{h(6Q_0 - 3\lambda t_2) + 3p\lambda}{t_1^2\lambda^3}, \\[2mm] c_2 = -\dfrac{-h\lambda^2 t_2^2 + 6h\lambda t_2 Q_0 - 6hQ_0^2 + 2p\lambda^2 t_2 - 6p\lambda Q_0 - 2K\lambda + 2h\lambda^2 t_1^2}{t_1^2\lambda^3}, \\[2mm] d_2 = -\dfrac{2p\lambda^2 t_2 Q_0 - 2hQ_0^3 - 3pQ_0^2\lambda - 2K\lambda Q_0 - hQ_0\lambda^2 t_2^2 + 3hQ_0^2\lambda t_2 - h\lambda^3 t_1^2 t_2 + hQ_0\lambda^2 t_1^2}{t_1^2\lambda^3}, \end{cases}$$

$A = b_2^2 - 3a_2c_2$, $B = b_2c_2 - 9a_2d_2$, $C = c_2^2 - 3b_2d_2$, $\Delta = B^2 - 4AC$, $\theta = \arccos(\frac{2Ab_2 - 3a_2B}{2\sqrt{A^3}})$, *and the optimal emergency order is made at* $t_e^* = \frac{Q_0 + Q_r^*}{\lambda}$ *with quantity* $Q_e^* = \lambda t_2 - Q_0 - Q_r^*$.

**Proof.** From the assumption and discussion in Section 3, the retailer's expected inventory cost is

$$\begin{aligned} G_2(Q_r = &-\frac{h(Q_0 + Q_r)^2(\lambda t_2 - Q_0 - Q_r)^2}{2t_1^2\lambda^3} + \frac{(p(\lambda t_2 - Q_0 - Q_1) - K)(Q_0 + Q_r)^2}{t_1^2\lambda^2} \\ &+ \frac{h(\lambda t_2 - Q_0 - Q_1)^2}{2\lambda} + \frac{h(Q_0^2 + Q_1^2)}{2\lambda} + 2K. \end{aligned}$$

To compute its minimum, we compute its derivative

$$G_2'(Q_r) = -\frac{1}{2t_1^2\lambda^3}\Big(2h(Q_0+Q_r)(\lambda t_2 - Q_0 - Q_r)^2 - 2h(Q_0+Q_r)^2(\lambda t_2 - Q_0 - Q_r)\Big)$$

$$+\frac{1}{t_1^2\lambda^2}\Big(-p(Q_0+Q_r)^2 + 2(p(\lambda t_2 - Q_0 - Q_r)-K)(Q_0+Q_r)\Big)$$

$$-\frac{h(\lambda t_2 - Q_0 - Q_r)}{\lambda} + \frac{hQ_r}{\lambda},$$

$$= -\frac{2h}{t_1^2\lambda^3}Q_r^3 - \frac{h(6Q_0 - 3\lambda t_2) + 3p\lambda}{t_1^2\lambda^3}Q_r^2$$

$$+\frac{-h\lambda^2 t_2^2 + 6h\lambda t_2 Q_0 - 6hQ_0^2 + 2p\lambda^2 t_2 - 6p\lambda Q_0 - 2K\lambda + 2h\lambda^2 t_1^2}{t_1^2\lambda^3}Q_r$$

$$+\frac{2p\lambda^2 t_2 Q_0 - 2hQ_0^3 - 3pQ_0^2\lambda - 2K\lambda Q_0 - hQ_0\lambda^2 t_2^2 + 3hQ_0^2\lambda t_2 - h\lambda^3 t_1^2 t_2 + hQ_0\lambda^2 t_1^2}{t_1^2\lambda^3}$$

$$= -(a_2 Q_r^3 + b_2 Q_r^2 + c_2 Q_r + d_2),$$

which is a cubic function of $Q_r$, where $a_2, b_2, c_2, d_2$ are given in the assertion. To solve the equation $G_2'(Q_r) = 0$, we break the discussion into four cases.

**Case 1.** $A = B = 0$. In this case, equation $G_2'(Q_r) = 0$ has one triple real root $Q_r = -\frac{b_2}{a_2}$, from which we conclude that $G_2(Q_r)$ is monotonically increasing in $(-\infty, -\frac{b_2}{a_2})$ and monotonically decreasing in $(-\frac{b_2}{a_2}, +\infty)$. So, $\frac{b_2}{a_2}$ is a local maximizer of the function $G_2(Q_r)$, and the minimum of $G_2(Q_r)$ in $[0, \lambda t_1 - Q_0]$ can be reached at 0 or $\lambda t_1 - Q_0$.

**Case 2.** $\Delta = B^2 - 4AC > 0$. In this case, equation $G_2'(Q_r) = 0$ has only one real root $\hat{Q}_r = \frac{-b_2 - (\sqrt[3]{Y_1} + \sqrt[3]{Y_2})}{3a_2}$, where $Y_1 = Ab_2 + \frac{3a_2}{2}\left(-B + \sqrt{B^2 - 4AC}\right)$ and $Y_2 = Ab_2 + \frac{3a_2}{2}\left(-B - \sqrt{B^2 - 4AC}\right)$. Then, the minimum of the $G_2(Q_r)$ in the $[0, \lambda t_1 - Q_0]$ can be reached in 0 or $\lambda t_1 - Q_0$.

**Case 3.** $\Delta = B^2 - 4AC = 0$. In this case, $G_2'(Q_r) = 0$ has two real roots $\hat{Q}_r = -\frac{b_2}{a_2} + \frac{B}{A}$, $\bar{Q}_r = -\frac{B}{2A}$. Then $G_2'(Q_r) = -\frac{2h}{t_1^2\lambda^3}(Q_r - \hat{Q}_r)(Q_r - \bar{Q}_r)^2$ from which we conclude that the minimum of the $G_2(Q_r)$ in the $[0, \lambda t_1 - Q_0]$ can be reached in 0 or $\lambda t_1 - Q_0$.

**Case 4.** $\Delta = B^2 - 4AC < 0$. In this case, the equation $G_2'(Q_r) = 0$ has three real roots $\hat{Q}_r = -\frac{1}{3a_2}(b_2 + 2\sqrt{A}\cos\frac{\theta}{3})$, $\bar{Q}_r = \frac{1}{3a_2}(-b + \sqrt{A}(\cos\frac{\theta}{3} - \sqrt{3}\sin\frac{\theta}{3}))$, $\tilde{Q}_r = \frac{1}{3a_2}(-b_2 + \sqrt{A}(\cos\frac{\theta}{3} + \sqrt{3}\sin\frac{\theta}{3}))$, where $\theta = \arccos(\frac{2Ab_2 - 3a_2 B}{2\sqrt{A^3}})$. Then,

$$G_2'(Q_r) = -\frac{2h}{t_1^2\lambda^3}(Q_r - \hat{Q}_r)(Q_r - \bar{Q}_r)(Q_r - \tilde{Q}_r),$$

from which we conclude that function $G_2(Q_r)$ is monotonically increasing in $(-\infty, \hat{Q}_r]$ and $[\bar{Q}_r, \tilde{Q}_r]$, and monotonically decreasing in $[\hat{Q}_r, \bar{Q}_r]$ and $[\tilde{Q}_r, +\infty)$. Then, $G_2(Q_r)$ in $[0, \lambda t_1 - Q_0]$ reaches the minimum at 0, $\bar{Q}_r$ or $\lambda t_1 - Q_0$.

From the discussions above, we can obtain the optimal regular order quantity

$$Q_r^* = \begin{cases} 0, & \text{if } \Delta < 0, \bar{Q}_r < 0, G_2(0) \le G_2(\lambda t_1 - Q_0); \\ \lambda t_1 - Q_0, & \text{if } \Delta < 0, \bar{Q}_r < 0, G_2(0) > G_2(\lambda t_1 - Q_0); \\ 0, & \text{if } \Delta < 0, 0 \le \bar{Q}_r \le \lambda t_1 - Q_0, G_2(0) \le \min(G_2(\bar{Q}_r), G_2(\lambda t_1 - Q_0)); \\ \bar{Q}_r, & \text{if } \Delta < 0, 0 \le \bar{Q}_r \le \lambda t_1 - Q_0, G_2(\bar{Q}_r) \le \min(G_2(0), G_2(\lambda t_1 - Q_0)); \\ \lambda t_1 - Q_0, & \text{if } \Delta < 0, \bar{Q}_r < 0, G_2(\lambda t_1 - Q_0) \le \min(G_2(0), G_2(\bar{Q}_r)); \\ 0, & \text{if } \Delta < 0, \bar{Q}_r > \lambda t_1 - Q_0, G_2(0) \le G_2(\lambda t_1 - Q_0); \\ \lambda t_0 - Q_0, & \text{if } \Delta < 0, \bar{Q}_r > \lambda t_1 - Q_0, G_2(0) > G_2(\lambda t_1 - Q_0); \\ 0, & \text{if } \Delta \ge 0, \quad G_2(0) \le G_2(\lambda t_1 - Q_0); \\ \lambda t_1 - Q_0. & \text{if } \Delta \ge 0, \quad G_2(0) > G_2(\lambda t_1 - Q_0). \end{cases}$$

From the discussion in Section 3, we know that $t_{e,2}^* = \frac{Q_0 + Q_r^*}{\lambda}$, and the optimal emergency order quantity $Q_e^* = \lambda t_2 - Q_0 - Q_r^*$. □

From conclusions obtained above, we can obtain Algorithm 2 for solving problem (2).

---

**Algorithm 2:** Solution method for optimization model (2).

---

**Step 1.** Input parameters $\lambda, K, h, p, Q_0, t_1, t_2$.

**Step 2.** Set

$$\begin{cases} a_1 = \dfrac{3h(\lambda t_2 - Q_0)}{t_1^2}, \quad c_1 = h(\lambda t_2 - Q_0), \\[2mm] b_1 = \dfrac{2p\lambda(\lambda t_2 - Q_0) - h(\lambda t_2 - Q_0)^2 - 2h(\lambda t_2 - Q_0)Q_0 - 2\lambda K}{\lambda t_1^2}. \end{cases}$$

If $b_1^2 - 4a_1 c_1 > 0$, set $\hat{t}_e = \frac{-b_1 + \sqrt{b_1^2 - 4a_1 c_1}}{2a_1}$, else set $\hat{t}_e = 0$.

**Step 3.** If $\hat{t}_e > \frac{Q_0}{\lambda}$, then $\hat{t}_e^* = \frac{Q_0}{\lambda}$, else $\hat{t}_e^* = \hat{t}_e$.

**Step 4.** Set

$$\begin{cases} a_2 = \dfrac{2h}{t_1^2 \lambda^3}, \quad b_2 = \dfrac{h(6Q_0 - 3\lambda t_2) + 3p\lambda}{t_1^2 \lambda^3}, \\[2mm] c_2 = -\dfrac{-h\lambda^2 t_2^2 + 6h\lambda t_2 Q_0 - 6hQ_0^2 + 2p\lambda^2 t_2 - 6p\lambda Q_0 - 2K\lambda + 2p\lambda^2 t_1^2}{t_1^2 \lambda^3}, \\[2mm] d_2 = -\dfrac{2p\lambda^2 t_2 Q_0 - 2hQ_0^3 - 3pQ_0^2 \lambda - 2K\lambda Q_0 - hQ_0 \lambda^2 t_2^2 + 3hQ_0^2 \lambda t_2 - h\lambda^3 t_1^2 t_2 + hQ_0 \lambda^2 t_1^2}{t_1^2 \lambda^3}, \end{cases}$$

and $A = b_2^2 - 3a_2 c_2$, $B = b_2 c_2 - 9a_2 d_2$, $C = c_2^2 - 3b_2 d_2$.

If $\Delta = B^2 - 4AC > 0$, set $Q_r = \frac{-b_2 - (\sqrt[3]{Y_1} + \sqrt[3]{Y_2})}{3a_2}$, where

$$\begin{cases} Y_1 = Ab_2 + \dfrac{3a_2}{2}\left(-B + \sqrt{B^2 - 4AC}\right), \\[2mm] Y_2 = Ab_2 + \dfrac{3a_2}{2}\left(-B - \sqrt{B^2 - 4AC}\right), \end{cases}$$

if $\Delta = B^2 - 4AC = 0$, set $Q_1 = -\frac{B}{2A}$, if $\Delta = B^2 - 4AC < 0$, set
$Q_r = \frac{1}{3a_2}(-b_2 + \sqrt{A}(\cos\frac{\theta}{3} - \sqrt{3}\sin\frac{\theta}{3}))$, where $\theta = \arccos(\frac{2Ab_2 - 3a_2 B}{2\sqrt{A^3}})$.

**Step 5.** If $Q_r < 0$ or $Q_r > \lambda t_1 - Q_0$, set $Q_r = 0$, else goto next step.

**Step 6.** If $G_2(Q_r) \leq min(G_2(\lambda t_1 - Q_0), G_2(0))$, then $Q_r^* = Q_1$, if
$G_2(0) \leq min(G_2(\lambda t_1 - Q_0), G_2(Q_r))$, then $Q_r^* = 0$, else $Q_r^* = \lambda t_1 - Q_0$.

**Step 7.** If $G_1(\hat{t}_e^*) \leq G_2(Q_r^*)$, then $t_e^* = \hat{t}_e^*$, else $t_e^* = \frac{Q_0 + Q_r^*}{\lambda}$.

**Step 8.** If $t_e^* \leq \frac{Q_0}{\lambda}$, then $Q_e^* = \lambda t_2 - Q_0$, else $Q_e^* = \lambda t_2 - Q_r^* - Q_0$.

**Step 9.** Output $t_e^*, Q_r^*, Q_e^*$.

---

From Theorems 3 and 4, we conclude that the algorithm can output a global optimal solution to problem (2). In the next section, we conduct numerical experiment to test the efficiency of the proposed algorithms.

## 5. Numerical Experiments

In this section, we first test the efficiency of the proposed model by two numerical examples and then make some numerical sensitivity analysis to test the influence of the involved parameters on the retailer's optimal ordering policy and retailer's expected inventory cost. In our numerical experiments, we use $\pi_1$ to denote the case that the happening time of the supply disruption obeys the uniform distribution $[0, t_1]$, and use $\pi_2$

to denote the case that the happening time of the supply disruption obeys the distribution with probability density function $f_2(x) = \frac{2x}{t_1^2}, x \in [0, t_1]$. Some data in our numerical examples are taken from [2,4].

**Example 1.** *Consider the inventory system with $t_1 = 15$, $t_2 = 60$, $Q_0 = 40$, $\lambda = 6$, $p = 80$, $h = 2$, $K = 20$, and the happening time of the supply chain disruption obeys the uniform distribution with density function $f_1(x) = \frac{1}{t_1}$ in $[0, t_1]$.*

By Algorithm 1, if the retailer makes an emergency order before $\frac{Q_0}{\lambda}$, then the retailer's optimal emergency ordering time is 4.18 and the optimal emergency order quantity is 320. Correspondingly, the retailer's expected inventory cost is 21,468. If the retailer chooses to make the emergency order after $\frac{Q_0}{\lambda}$, then a regular order with quantity 50 is needed, and the optimal emergency ordering time is 15, the optimal emergency ordering quantity is 270. Correspondingly, the retailer's expected inventory cost is 22,308. Thus, the optimal emergency ordering strategy is making the emergency order at 4.18 with quantity 320. The detailed numerical result for this example is listed in Table 2, where we use Strategy I to denote the emergency procurement strategy that makes one emergency order before $\frac{Q_0}{\lambda}$, and use Strategy II to denote the emergency procurement strategy that makes one regular order at $\frac{Q_0}{\lambda}$ and an emergency order later.

**Table 2.** Numerical result for Example 1.

| Policy | $t_e^*$ | $Q_r^*$ | $Q_e^*$ | $G_1(t_e^*)$ | $G_2(Q_r^*)$ |
|--------|---------|---------|---------|--------------|--------------|
| I | 4.18 | / | 320 | 21,468 | / |
| II | 15 | 50 | 270 | / | 22,308 |

**Example 2.** *For the inventory system considered in Example 1, suppose the happening time of the supply chain disruption obeys the distribution with probability density function $f_2(x) = \frac{2x}{t_1^2}$ in $[0, t_1]$ and other parameters remain unchanged.*

By Algorithm 2, if the retailer makes an emergency order before $\frac{Q_0}{\lambda}$, then the emergency order is made at the beginning with a quantity of 320. Correspondingly, the retailer's expected cost is 20,935. If the retailer chooses to make an emergency order after $\frac{Q_0}{\lambda}$, then the retailer should make a regular order with quantity 3.3 first, then make an emergency order with quantity 316.6 at 7.2. Under this strategy, the retailer's expected inventory cost is 19,015. Thus, the retailer should make an emergency order after $\frac{Q_0}{\lambda}$. The detailed numerical result for this example is listed in Table 3.

**Table 3.** Numerical result for Example 2.

| Policy | $t_e^*$ | $Q_r^*$ | $Q_e^*$ | $G_1(t_e^*)$ | $G_2(Q_r^*)$ |
|--------|---------|---------|---------|--------------|--------------|
| I | 0 | / | 320 | 20,935 | / |
| II | 7.2 | 3.3 | 316.6 | / | 19,015 |

Now, we make some sensibility analysis of the demand rate $\lambda$ on the optimal emergency ordering policy. For the inventory mechanism of Example 1, we let $\lambda$ increase from 5 to 7, while keeping other parameters unchanged. The numerical results are shown in Figures 3–6, from which we can see that with the increase of the demand rate $\lambda$, the ordering time of the emergency order becomes earlier. Compared with the case that the happening time of the supply disruption obeys distribution $\pi_1$, the demand rate $\lambda$ has a larger influence on the ordering time of the emergency order for the case that the happening time of the supply disruption obeys distribution $\pi_2$.

Now, we make some sensibility analysis of the unit holding cost $h$ on the emergency ordering policy. For the inventory mechanism considered in Example 1, we let $h$ increases from 1.5 to 2.5, while keeping other parameters unchanged. The numerical results are shown in Figures 7 and 8, from which we can see that when $h$ increases, the ordering time of the emergency order becomes earlier. Compared with the case where the happening time of the supply disruption obeys distribution $\pi_2$, the parameter $h$ has a complex effect on the retailer's optimal ordering policy for the case that the happening time of the supply disruption obeys distribution $\pi_1$, see Figure 9 and Figure 10.

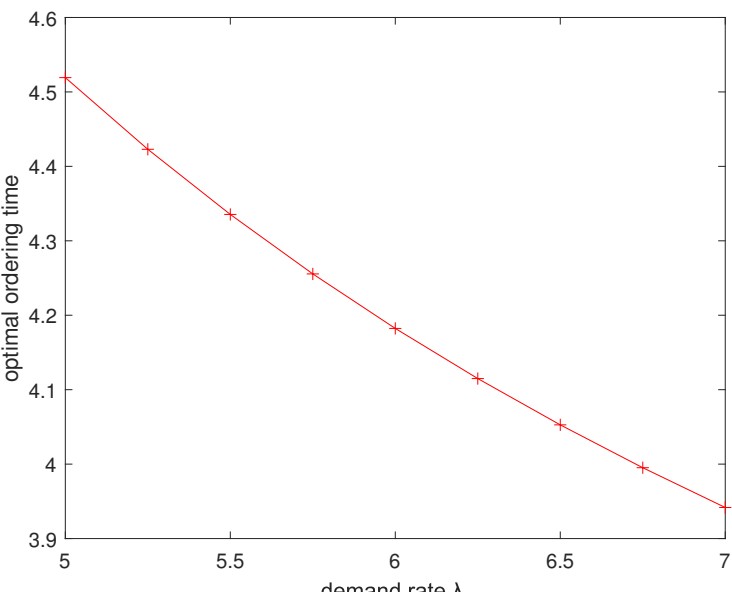

**Figure 3.** Influence of $\lambda$ on emergency ordering time for distribution $\pi_1$.

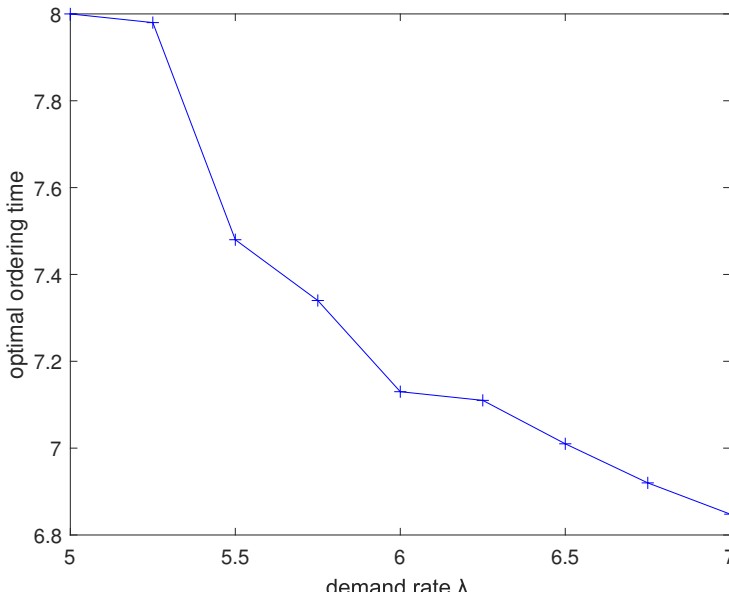

**Figure 4.** Influence of $\lambda$ on emergency ordering time for distribution $\pi_2$.

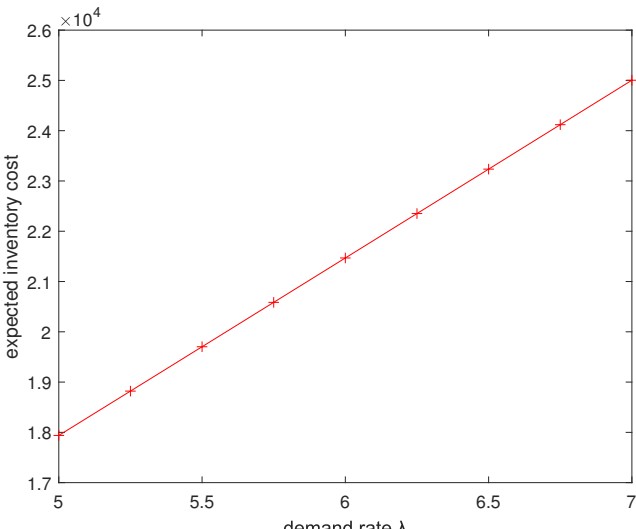

**Figure 5.** Influence of $\lambda$ on inventory cost for distribution $\pi_1$.

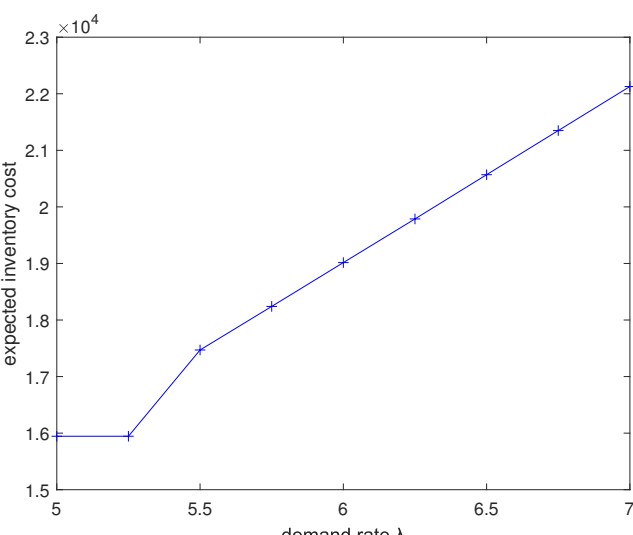

**Figure 6.** Influence of $\lambda$ on inventory cost for distribution $\pi_2$.

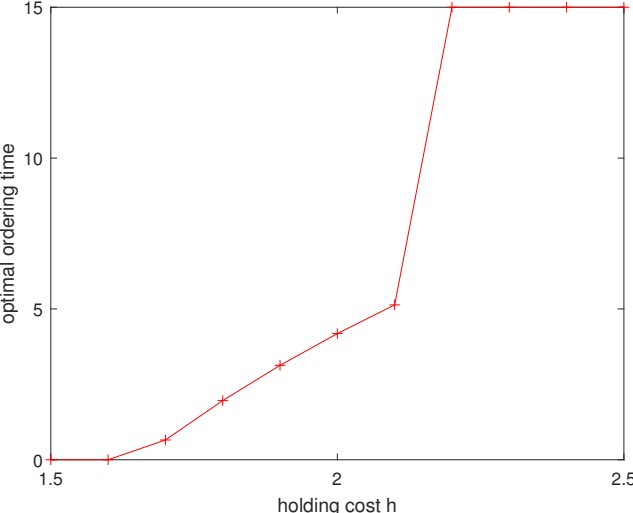

**Figure 7.** Influence of $h$ on emergency ordering time for distribution $\pi_1$.

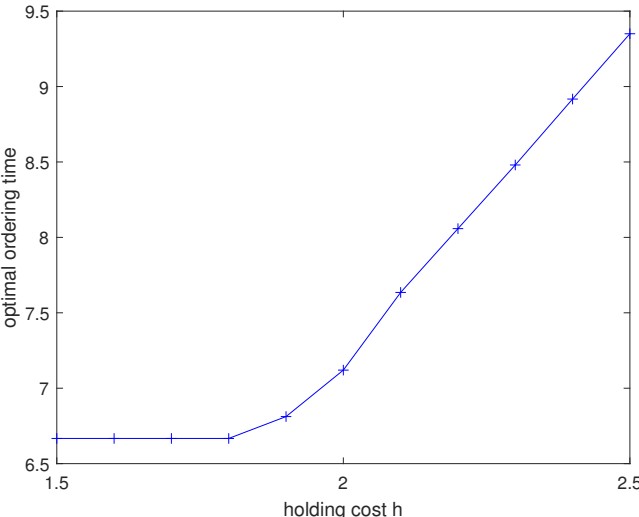

**Figure 8.** Influence of $h$ on emergency ordering time for distribution $\pi_2$.

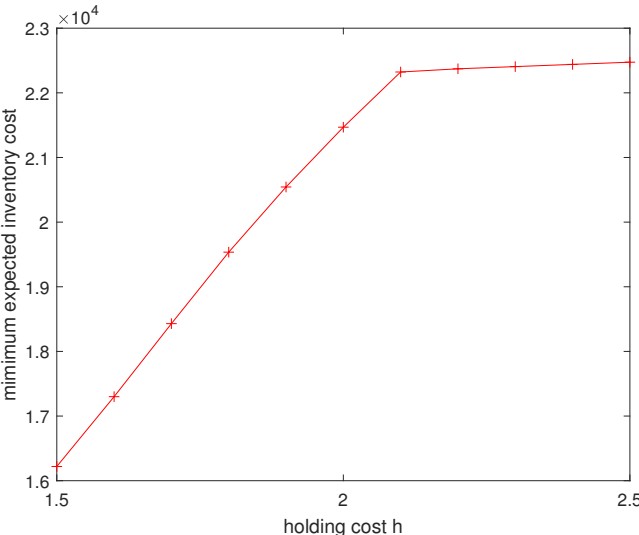

**Figure 9.** Influence of $h$ on inventory cost for distribution $\pi_1$.

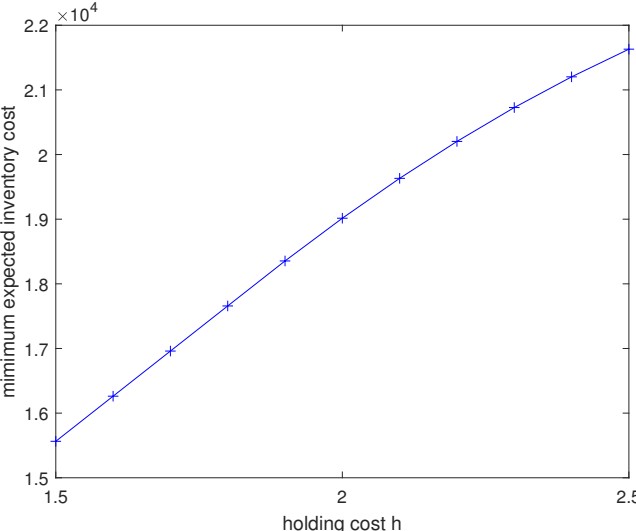

**Figure 10.** Influence of $h$ on inventory cost for distribution $\pi_2$.

Now, we make some sensibility analysis of the unit holding cost $p$ on the emergency ordering policy. For the inventory mechanism considered in Example 1, we let $p$ increase from 78 to 82, while keeping other parameters unchanged. The numerical results are shown in Figures 11–14, from which we can see that when $p$ increases, the emergency order is made earlier. Compared with the case where the happening time of the supply disruption obeys distribution $\pi_1$, the parameter $p$ has a complex effect on the retailer's optimal ordering policy for the case where the happening time of the supply disruption obeys distribution $\pi_2$.

From the numerical experiments on the involved parameter sensitivity analysis, we can see that the variation of holding cost $h$, demand rate $\lambda$, and deadline time of the supply disruption $t_1$ have a larger influence on the emergency ordering policy, while the stockout penalty $p$ has less influence on the emergency ordering policy. This means that the holding cost $h$, demand rate $\lambda$, and deadline time of the supply disruption $t_1$ are crucial ingredients in the emergency procurement problem and the manager should pay more attention to them.

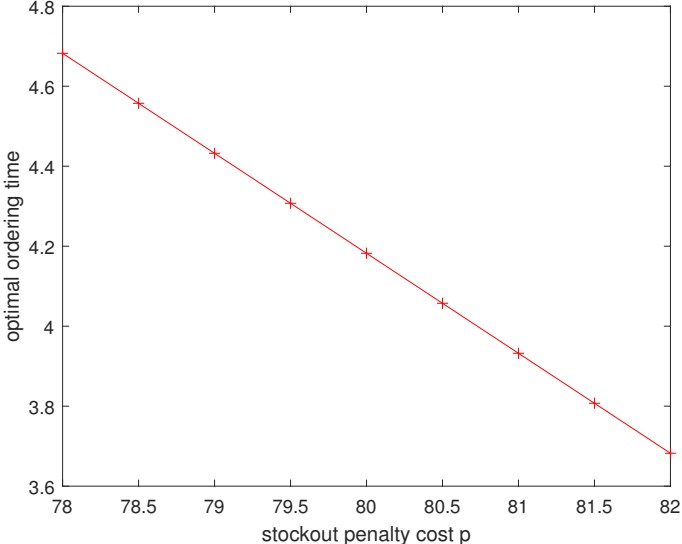

**Figure 11.** Influence of $p$ on emergency ordering time for distribution $\pi_1$.

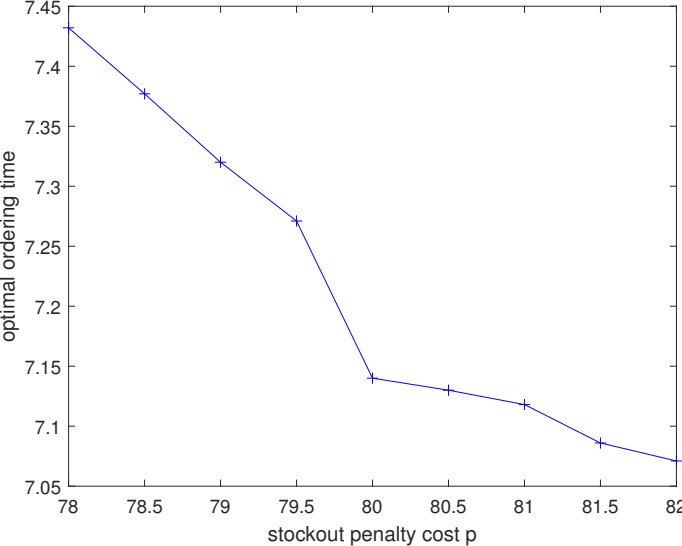

**Figure 12.** Influence of $p$ on emergency ordering time for distribution $\pi_2$.

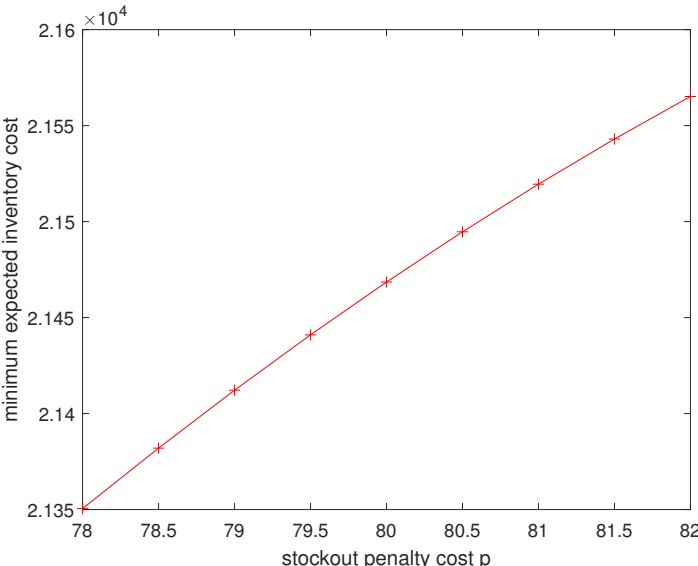

**Figure 13.** Influence of $p$ on inventory cost for distribution $\pi_1$.

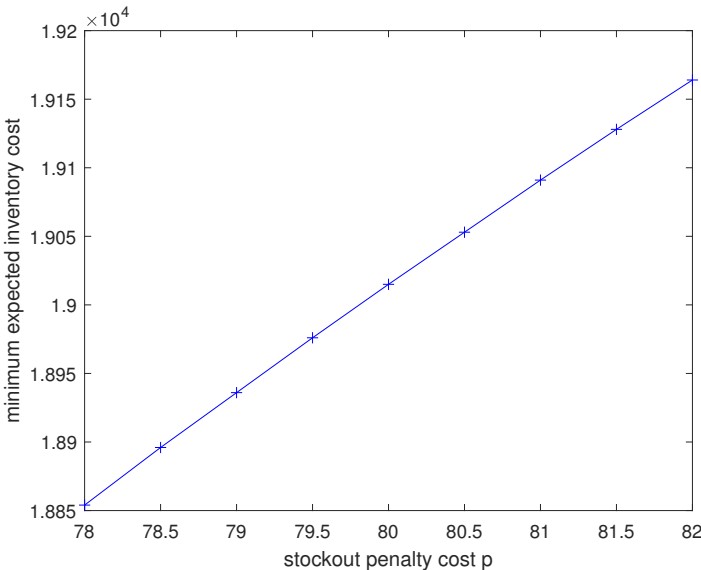

**Figure 14.** Influence of $p$ on inventory cost for distribution $\pi_2$.

Finally, we test the influence of the deadline time of the supply disruption on the emergency procurement strategy.

**Example 3.** *Consider the inventory system with* $t_2 = 60$, $Q_0 = 40$, $\lambda = 6$, $p = 80$, $h = 2$, $K = 20$; *we let* $t_1$ *increases from* 13 *to* 20.

For this inventory mechanism, for the case that the happening time of supply disruption obeys distribution $\pi_1$, the numerical results are shown in Figure 15, from which we can see that when $t_1$ increases from 13 to 18, the optimal emergency procurement strategy is Strategy I, and when $t_1$ increases from 19 to 20, the optimal emergency procurement strategy is Strategy II.

For the case that the happening time of the supply disruption obeys distribution $\pi_2$, the numerical results are shown in Figure 16, from which we can see that when $t_1$ increases 13 to 20, the optimal emergency procurement strategy is always Strategy II.

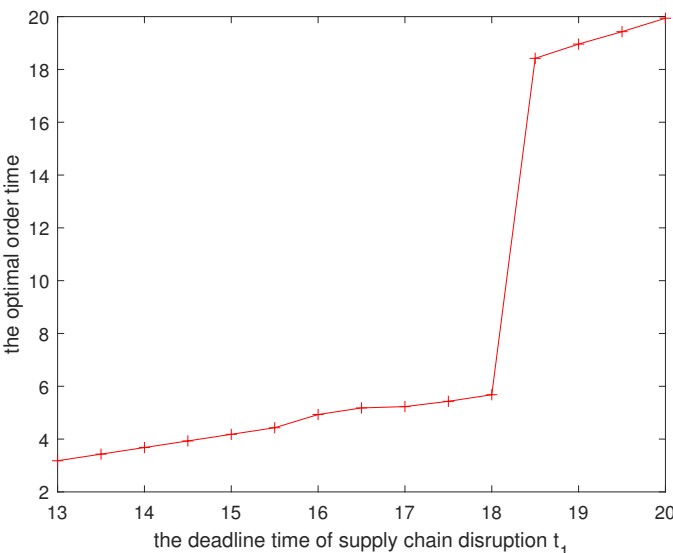

**Figure 15.** Influence of $t_1$ on emergency ordering strategy for distribution $\pi_1$.

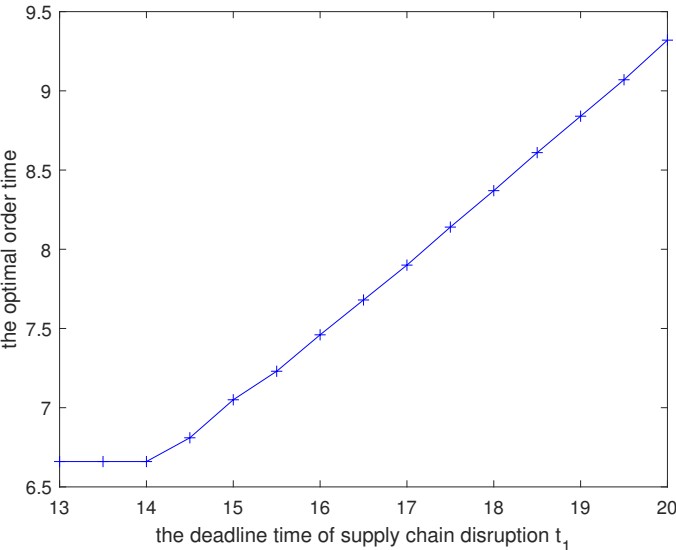

**Figure 16.** Influence of $t_1$ on emergency ordering strategy for distribution $\pi_2$.

From the numerical experiments, we can see that the probability distribution of the happening time of the supply disruption has a larger influence on the emergency procurement strategy. Thus, the probability distribution of the happening time of the supply disruption is also a crucial ingredient in the emergency procurement problem.

## 6. Conclusions

This paper considered the emergency procurement problem with an impending supply disruption, which can be encountered in reality. To solve the problem, we formulate it as an optimization model based on minimizing the inventory cost, and by the model analysis, we gave an optimal emergency procurement policy to the retailer. Some numerical experiments are provided, which give some useful suggestions to the retailer when facing an impending supply disruption.

Certainly, the emergency procurement problem considered in this paper assumes that happening time of the supply disruption obeys two common probability distributions and the ending time of the event is deterministic. However, in many cases, the assumptions

can not be fulfilled. Thus, one extension of research is that the happening time of the event obeys a more practical probability distribution and the ending time of the event is nondeterministic. Another extension of the research is to take the retailer's risk preference into consideration, which can make the research more significant.

**Author Contributions:** Conceptualization, J.H., Y.W.; methodology, J.H., G.W; software, J.H, Y.W.; writing—original draft preparation, J.H., G.W and Y.W.; writing—review and editing, J.H., G.W. and Y.W.; supervision, Y.W.; funding acquisition, Y.W. All authors have read and agreed to the published version of the manuscript.

**Funding:** This research is supported by the National Natural Science Foundation of China (No.12071250).

**Institutional Review Board Statement:** Not applicable.

**Informed Consent Statement:** Not applicable.

**Acknowledgments:** The authors are in debt to three anonymous referees for numerous insightful comments and suggestions, which have greatly improved the paper.

**Conflicts of Interest:** The authors declare no conflict of interest.

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
