# Peer review of "Retailer’s Emergency Ordering Policy When Facing an Impending Supply Disruption"

_sustainability, doi:10.3390/su13137041_

Round 1

Reviewer 1 Report

Thank you for the opportunity to review this manuscript. While I do potentially see value in the work that the authors have done, there are some significant changes that must be made to the manuscript in order to justify not only the model set up, but also the author's discussion and interpretation of their model results.

In terms of model set up, there are a couple of issues that need to be addressed. The biggest issue that I see is that the scenario proposed by the authors, as well as the assumptions suggested by their model, don't really align with the reality of decision making during supply chain disruptions. For example, the authors give the disruption a uniform probability distribution of [0,to] for the timing of a disruption event where the disruption event's last possible occurrence date is known to the decision maker. This is highly irregular; very rarely would any disruption have a known date by which it must occur, and if that were the case then it would be rather hard to classify the occurrence as a disruption since a significant amount of forward planning is possible in response to this (as the results of the model show). The authors need to devote more time early in the manuscript to justifying why this look at a disruption is valid; some discussion of other published papers that have made this assumption would be most helpful in regard. I understand that the results of the model are predicated on this assumption, so it needs to be validated as an assumption that the field of supply chain disruption makes on occasion.

As a secondary concern to the model set up, the authors should revise their subscripts to be a little bit more consistent in terms of meaning; a good example of this is the to and Qo variables. to represents the last possible moment that the disruption can occur, while Qo represents the firm's inventory level when the disruption occurs. The disruption could occur well before to, meaning that these subscripts aren't necessarily representative of the same thing. However, td and Qd do match up. While it's only a stylistic thing, it would help improve the clarity of the model and allow the reader to more easily relate variables by the moment in time they are determined.

Also, related to case II: how can a firm make a regular order and an emergency order during a disruption? Isn't the whole point of a disruption the fact that it forces the retailer into behavior that is not normal? How would this work mechanically?

Lastly, for the sensitivity analysis, how did you go about choosing the sensitivities of the parameters? You vary the demand rate from 5 to 7, why? Is there something about that demand rate that lines up with a realistic scenario? How about the holding costs? Do these reflect realistic scenarios? There just needs to be some type of justification for why these are the sensitivities chosen for these parameters.

Overall, there is potential in this piece, but a stronger justification for the model set up and assumptions made are key missing components that need to be addressed.

Author Response

First, we thank you for your constructive comments which help to improve the paper. The manuscript was revised completely according to the referee's reports. The details for replies are given below. 1. In terms of model set up, there are a couple of issues that need to be addressed. The biggest issue that I see is that the scenario proposed by the authors, as well as the assumptions suggested by their model, don't really align with the reality of decision making during supply chain disruptions. For example, the authors give the disruption a uniform probability distribution of [0,to] for the timing of a disruption event where the disruption event's last possible occurrence date is known to the decision maker. This is highly irregular; very rarely would any disruption have a known date by which it must occur, and if that were the case then it would be rather hard to classify the occurrence as a disruption since a significant amount of forward planning is possible in response to this (as the results of the model show). The authors need to devote more time early in the manuscript to justifying why this look at a disruption is valid; some discussion of other published papers that have made this assumption would be most helpful in regard. I understand that the results of the model are predicated on this assumption, so it needs to be validated as an assumption that the field of supply chain disruption makes on occasion. REPLY: For the set up of the model and the assumption on the model, we add some explanation in the third to the last paragraph of Section 1. 2. As a secondary concern to the model set up, the authors should revise their subscripts to be a little bit more consistent in terms of meaning; a good example of this is the to and Qo variables. to represents the last possible moment that the disruption can occur, while Qo represents the firm's inventory level when the disruption occurs. The disruption could occur well before to, meaning that these subscripts aren't necessarily representative of the same thing. However, td and Qd do match up. While it's only a stylistic thing, it would help improve the clarity of the model and allow the reader to more easily relate variables by the moment in time they are determined. REPLY: For the symbols used in the discussion of the model, we have modified them, see Figs 1.1 and 1.2, and Table 1.1. 3. Also, related to case II: how can a firm make a regular order and an emergency order during a disruption? Isn't the whole point of a disruption the fact that it forces the retailer into behavior that is not normal? How would this work mechanically? REPLY: An explanation on the strategy with a regular order is given in the first paragraph of Section 2. Lastly, for the sensitivity analysis, how did you go about choosing the sensitivities of the parameters? You vary the demand rate from 5 to 7, why? Is there something about that demand rate that lines up with a realistic scenario? How about the holding costs? Do these reflect realistic scenarios? There just needs to be some type of justification for why these are the sensitivities chosen for these parameters. REPLY: This was explained in the last paragraph of Section 5.

Reviewer 2 Report

The paper is very nice, engaging, and well decorated. However, the following comments should be cleared during revisions.

  1. The abstract and conclusions are very short.  The abstract should contain the details of the study and the findings in a very constructive way. The abbreviation should not be in the abstract. If needed, it can be started from the introduction onwards. The conclusions should be extended with significant findings and limitations. The applicability of the model should be explained.
  2. The research gap should be adequately explained.
  3. In the introduction, please rearrange/rewrite so that each authors’/most of the authors' contributions should be linked. Please try to maintain the literature sequentially.
  4. Please write proper managerial insights to show the industry managers' benefit from this research (See the study, cite them, and write this way: Optimum ordering policy for an imperfect single-stage manufacturing system with safety stock and planned backorder; Sustainable ordering policies for non-instantaneous deteriorating items under carbon emission and multi-trade-credit-policies; Ordering and transfer policy and variable deterioration for a warehouse model).
  5. Please write the significant findings in conclusions. Do not mention all assumptions which have been indicated within the model.
  6. Each function should be adequately explained (See the study, cite them, and write this way: Synergic effect of reworking for imperfect quality items with the integration of multi-period delay-in-payment and partial backordering in global supply chains; A Sustainable Online-to-Offline (O2O) Retailing Strategy for a Supply Chain Management under Controllable Lead Time and Variable Demand).
  7. Explain the significant findings. What are the managerial insights?
  8. Is the optimal value the global optimum?
  9. What is the data source of the numerical experiment? Please mention that the data is from industry or some literature, i.e., accurate data or artificial data.
  10. Make the graphical representation properly for the research from the sensitivity analysis table.

Author Response

First, we thank you for your constructive comments which help to improve the paper. The manuscript was revised completely according to the referee's reports. The details for replies are given below.

1. The abstract and conclusions are very short.The abstract should contain the details of the study and the findings in a very constructive way. The abbreviation should not be in the abstract. If needed, it can be started from the introduction onwards. The conclusions should be extended with significant findings and limitations. The applicability of the model should be explained.

REPLY: This has been done as suggested. See the bastract and conclusions in the revised version.

2. The research gap should be adequately explained.

REPLY: This was done in the conclusion section.

3. In the introduction, please rearrange/rewrite so that each authors’/most of the authors' contributions should be linked. Please try to maintain the literature sequentially.

REPLY: This was done as suugested, see the revised version.

4. Please write proper managerial insights to show the industry managers' benefit from this research (See the study, cite them, and write this way: Optimum ordering policy for an imperfect single-stage manufacturing system with safety stock and planned backorder; Sustainable ordering policies for non-instantaneous deteriorating items under carbon emission and multi-trade-credit- policies; Ordering and transfer policy and variable deterioration for a warehouse model).

REPLY: This was done, see the references in the new version.

5. Please write the significant findings in conclusions. Do not mention all assumptions which have been indicated within the model.

REPLY: This was done as suggested.

6. Each function should be adequately explained (See the study, cite them, and write this way: Synergic effect of reworking for imperfect quality items with the integration of multi-period delay-in- payment and partial backordering in global supply chains; A Sustainable Online-to-Offline(O2O) Retailing Strategy for a Supply Chain Management under Controllable Lead Time and Variable Demand).

REPLY: This was done as suggested.

7.Explain the significant findings. What are the managerial insights?

REPLY: This was done in the conclusion section.

8. Is the optimal value the global optimum?

REPLY: Yes, this was confirmed after the description of each algorithms.

  1. What is the data source of the numerical experiment? Please mention that the data is from industry or some literature, i.e., accurate data or artificial data.

The data of the numerical experiment is artificial according to some literatures.

  • Make the graphical representation properly for the research from the sensitivity analysis table.

This was done in the end of Section 5.

Reviewer 3 Report

Retailer’s ordering policy for inventory mechanism with a supply chain disruption

numerical experiments are not enough to test a validity of something in science

Line 46

In this paper, we consider the inventory mechanism with a stable demand and with

a supply disruption whose happening time is random and ending time is deterministic.

What actual situation has a deterministic end time worth researching? Give examples.

With random disruption of an event how can someone prepare and buy in advance. There are also language problems in that sentence

Line 66

Could you be more specific – which probability distribution did you use

Table 1 – what is the difference between these two notions?

t0 the deadline happening time emergency of the supply interruption

te the ending time of supply interruption

Line 96

If 95 (3) the retailer only makes one emergency order before t0.  What is than ts the happening time of the supply disruption

Line 113

density function f2(x) = 2x/t20

Please could you explain why you used this density function – there has to be arguments why something is used.

Author Response

First, we thank you for your constructive comments which help to improve the paper. The manuscript was revised completely according to the referee's reports. The details for replies are given below.

1. Line 46 In this paper, we consider the inventory mechanism with a stable demand and with a supply disruption whose happening time is random and ending time is deterministic.What actual situation has a deterministic end time worth researching? Give examples.

REPLY: An example is given in the last part of Section 1.

2. With random disruption of an event how can someone prepare and buy in advance.

REPLY: This was explained in the first paragraph of Section 2.

3. There are also language problems in that sentence. Line 66 Could you be more specific – which probability distribution did you use Table 1 – what is the difference between these two notions? t0 the deadline happening time emergency of the supply interruption te the ending time of supply interruption

REPLY: This was revised as suggested, see Table 2.1

4. Line 96 If 95 (3) the retailer only makes one emergency order before t0. What is than ts the happening time of the supply disruption. The happening time of the supply disruption is random, Line 113 density function f2(x) = 2x/t20.Please could you explain why you used this density function – there has to be arguments why something is used.

REPLY: This was explained after the assumption given in the end of Section 2.

Round 2

Reviewer 1 Report

My comments have largely been addressed; the only remaining issue is that the scenario for the numerical experiments is still presented without sufficient reference to a realistic event or scenario that it would depict, but that's not a fatal flaw. I would recommend the authors bring their explanation for the use of these specific values to the front end of section 5.

Author Response

The only remaining issue is that the scenario for the numerical experiments is still presented without sufficient reference to a realistic event or scenario that it would depict, but that's not a fatal flaw.

REPLY: This has been revised as suggested, see the end of the first paragraph of Section 5.

I would recommend the authors bring their explanation for the use of these specific values to the front end of Section 5.

REPLY: This has been done in the first paragraph of Section 5.

Reviewer 2 Report

The paper just double check the reference citations. Rests are okay. Please proceed for acceptance with correct reference citation.

Author Response

The paper just double check the reference citations. Resets are okay. Please proceed for acceptance with correct reference citation.

REPLY: This has been done as suggested. 

Reviewer 3 Report

The authors stick to their unrealistic model even though both reviewers commented on that. They say that they give examples but actually they don't. Not much effort is put to resolve issues of both reviewers.

Why are question marks instead of references on the first page?

Author Response

The authors stick to their unrealistic model even though both reviewers commented on that. They say that they give examples but actually they don't. Not much effort is put to resolve issues of both reviewers.

REPLY: A realsitic example on the concerned model is given in the third paragraph to the last in Section 1. 

Why are question marks instead of references on the first page?

REPLY: This has been revised.

Round 3

Reviewer 3 Report

Ok, authors after insisting on real world examples did put some in. In the conclusion they put limitation that such events are rare.